# The Effect of Nutritional Intervention in Nutritional Risk Screening on Hospitalised Lung Cancer Patients

**DOI:** 10.3390/nu17010006

**Published:** 2024-12-24

**Authors:** Raquel Oliveira, Bruno Cabrita, Ângela Cunha, Sónia Silva, João P. M. Lima, Diana Martins, Fernando Mendes

**Affiliations:** 1Pulmonology Service, Centro Hospitalar de Leira, Rua das Olhalvas, 2414-016 Leiria, Portugal; raquel.oliveira@ulsrl.min-saude.pt (R.O.); bruno.cabrita@chleiria.min-saude.pt (B.C.); angela.cunha@chleiria.min-saude.pt (Â.C.); sonia.silva@ulsrl.min-saude.pt (S.S.); 2ciTechCare—Center for Innovative Care and Health Technology, Health Innovation Hub|Politécnico de Leiria, Campus 5, Rua das Olhalvas, 2414-016 Leiria, Portugal; 3Higher School of Health, University of Algarve I University of Algarve—Campus de Gambelas, 8005-139 Faro, Portugal; 4H&TRC—Health & Technology Research Center, Coimbra Health School, Polytechnic University of Coimbra, 3045-043 Coimbra, Portugal; joao.lima@estesc.ipc.pt (J.P.M.L.); diana.martins@estesc.ipc.pt (D.M.); 5Coimbra Health School (ESTeSC), Polytechnic University of Coimbra, 3046-854 Coimbra, Portugal; 6Biophysics Institute of Faculty of Medicine, Coimbra Institute for Clinical and Biomedical Research (iCBR) Area of Environment Genetics and Oncobiology (CIMAGO), University of Coimbra, 3000-548 Coimbra, Portugal; 7Center for Innovative Biomedicine and Biotechnology, University of Coimbra, 3000-548 Coimbra, Portugal; 8European Association for Professions in Biomedical Sciences, 1000 Brussels, Belgium

**Keywords:** nutrition assessment, nutrition therapy, lung cancer

## Abstract

Background: Lung cancer (LC) patients are prone to suffer from malnutrition. Malnutrition negatively affects patients’ response to therapy, increases the incidence of treatment-related side effects, and decreases survival. Early identification of LC patients who are malnourished or at risk of malnutrition can promote recovery and improve prognosis. Objective: This study aimed to assess the risk and nutritional status of lung cancer patients who are hospitalised, as well as to evaluate the impact of nutritional intervention on the risk of malnutrition. Methods: From January 2022 to December 2023, 53 LC patients hospitalised in a pulmonology department had their nutritional risk (initial and final) and nutritional status (initial) assessed. All were selected for nutritional intervention. Nutrition counselling was the first intervention option, along with dietary changes with/without oral nutritional supplements. Results: At the time of hospitalisation, 90.6% of the patients were at nutritional risk, 45.3% were classified as moderately malnourished, and 35.8% were classified as severely underweight. After the hospitalisation, 73.6% were at nutritional risk at the time of discharge, suggesting a statistically significant decrease in the number of patients with nutritional risk. Conclusions: Most LC patients hospitalised presented an altered nutritional status. Our study suggests that a nutritional intervention must be implemented to reduce malnutrition risk, which may impact prognosis. The comprehensive nutritional problems experienced by LC patients require nutritional assessment and improved individually tailored nutritional support.

## 1. Introduction

Lung cancer (LC) is the most commonly diagnosed cancer worldwide (12.4% of the total cases), followed by cancers of the female breast (11.6%), colorectum (9.6%), prostate (7.3%), and stomach (4.9%). LC is also the leading cause of cancer death (18.7% of the total cancer deaths, over 1.8 million deaths worldwide) [1]. The disease is the leading cause of cancer death among men. The disease is second among women for both incidence and mortality, with male-to-female ratios for lung cancer incidence and mortality approximately 2:1. However, these ratios differ significantly by region, ranging from nearly equal in North America and Northern Europe to four-to-five-times higher in Northern Africa and Eastern Europe [1]. In Europe, LC is the leading cause of cancer mortality among men in all countries except Sweden and among women in 13 countries (one-third of the European nations). The disease has a more significant impact on men than on women, with a male-to-female incidence ratio varying from 3 to 10 [2]. Prostate (21%), colorectal (20%), and lung (12%) cancers are the most common cancers in men in Portugal, while in women breast cancer is the cancer with the highest incidence (28%), followed by colorectal cancer (16%) and LC (6%) [2]. LC is the leading cause of cancer mortality among men in Portugal and the third leading cause among women [2].

Depending on the disease stage and treatment modality, patients with LC are prone to malnutrition. Malnutrition negatively affects patients’ response to therapy, increases the incidence of treatment-related side effects, and can contribute to a survival decrease. Malnutrition is a comorbidity frequently found in neoplastic patients, but it remains often underestimated and thus undertreated [3]. Early identification of patients who are malnourished or at risk of malnutrition can promote recovery and improve prognosis. In addition, early nutritional intervention is cost-effective, as it reduces complication rates and length of hospitalisation [4]. Malnutrition in people with LC is highly prevalent (35–70%) depending on the treatment type, stage of disease, and assessment method [4,5]. LC is one of the most common cancers worldwide; however, although malnutrition is frequent in LC patients, this aspect is underestimated and thus undertreated [6]. Malnutrition screening is the starting point for high-quality nutrition care. The systematic nutritional risk assessment in Portugal is mandatory for every hospital inpatient using the Nutritional Risk Screening 2002 [7,8].

Nutritional intervention is mandatory as an adjuvant to any treatment, as it improves nutritional parameters, body composition, symptoms, quality of life, and, ultimately, survival [3,8]. Risk factors for malnutrition have been described, such as nutritional deficiencies resulting from the disease per se, such as anorexia and nausea, and others resulting from therapeutic procedures during or before hospitalisation. Precipitating and/or aggravating causes of malnutrition during hospitalisation can be considered in situations such as the use of serums, prolonged fasting, skipping meals for tests, lack of control over intake, failure to record weight and its evolution, and delays in starting nutritional support [4,9].

Malnutrition changes the immunological, cardiovascular, respiratory, and gastrointestinal systems. These changes result in complications such as deterioration in physical and mental capacity, increased susceptibility to infections, and delayed wound healing, with the inherent risk of an increased length of hospital stay and morbidity and mortality [4,10,11,12]. Nutritional status is a significant clinical and prognostic indicator in evaluating lung cancer treatment. Malnutrition is linked to a poorer outcome in terms of overall survival, time to tumour progression, and quality of life in patients undergoing treatment for lung cancer [13].

All individuals previously identified as being at nutritional risk during screening should have their nutritional status assessed. This assessment is part of this comprehensive study, and good-quality care is given to hospitalised patients [10,11,12]. The purpose of evaluating nutritional status is to detect malnourished patients and their degree of malnutrition, identify patients who need nutritional support, program an appropriate dietary plan, and assess the effectiveness of the nutritional support provided as part of the hospital treatment interventions [11,14].

It has been shown that the prevention and early treatment of malnutrition improves the quality of care in both clinical and economic terms. All strategies should be used to improve the nutritional status of hospitalised patients [12,15]. Several studies concluded that introducing nutritional support for malnourished patients significantly reduces hospitalisation time. Even though there are costs associated with the nutritional backing, these are offset by savings in hospitalisation time [3,15,16,17].

Insufficient food intake increases the prevalence and severity of malnutrition, with a concomitant increase in its complications. Therefore, all hospitalised patients must have a personalised nutritional care plan [8,9]. Dietary approaches, with a view to immediate nutritional intervention appropriate to the clinical situation, can take various forms, such as an oral diet enriched with nutritionally dense foods, diet changes (texture changes, inclusion of small intermediate meals), or even artificial oral feeding, enteral nutrition, or parenteral nutrition [16]. Dietary counselling should also be implemented as one of the first measures, and the patient should be involved in the nutritional care plan [11,14].

Many LC inpatients depend on hospital food, so it should be considered an integral part of individualised treatment, not just hospital routine. This means that assessing hospital food intake plays a vital role during hospitalisation. A patient with an adequate dietary intake will have a lower nutritional risk. ‘Healthy’ diets low in fats and sugars may be necessary for patients with coronary pathology or obesity, but most patients need diets fortified with energy and nutrients. Some actions should be implemented to change this scenario: food availability for intermediate meals and supplements can be used but should never be a substitute for regular food [17,18]. Once nutritional support has been implemented, monitoring is essential to assess its effectiveness and allow eventual corrections according to the patient’s needs. Different interventions may be necessary during hospitalisation. Monitoring should include food intake, nutritional intervention tolerance, weight change, changes in laboratory parameters, etc. [8,12,18,19,20,21].

We aimed to assess the risk and nutritional status of hospitalised LC patients and evaluate the impact of nutritional intervention on the risk of malnutrition.

## 2. Materials and Methods

### 2.1. Type of Study

A prospective observational study was conducted in patients hospitalised with LC or lung metastasis, who were offered nutritional evaluation and support. This study was conducted in the Centro Hospitalar de Leiria between January 2022 and December 2023. Only patients with LC as a primary tumour or patients with lung metastasis (another primary tumour) were selected for this study. The exclusion criteria include patients with less than 18 years of age and patients with pulmonary nodules or consolidation that were not attributable to cancer.

The oncology treatment was intended to be provided according to national and international guidelines.

### 2.2. Data Collection

All the subjects were interviewed by the same two leading investigators responsible for this study. The collected data included a detailed clinical history and subjective and objective examinations. Biometric parameters were also collected to facilitate the Nutritional Risk Screening 2002 (NRS2002) calculation, at both admission and discharge, to evaluate nutritional risk. Nutritional status was assessed according to the Global Leadership Initiative on Malnutrition (GLIM) criteria [22,23,24].

Anthropometric measurements were taken according to the service’s standard procedure:

Patients were weighed on the same scale (Seca 664 electronic wheelchair scale(Seca, CA, USA) under the same conditions. When it was not possible to weigh the patient, the weight reported by the patient or their family member or estimated by those responsible for this study was used (using the formula of Chumeal et al. (1988) [25]).

At the end of hospitalisation, the patients were weighed again, and the difference in weight compared to that recorded at the beginning of hospitalisation was calculated (except for patients who died). The height value reported by the patients or their relatives was used.

Data on food intake were collected from the patient’s medical file. Food intake was assessed and recorded by the nursing team as a percentage of intake (0%, 25%, 50%, 75%, or 100%) in accordance with the service’s usual procedure.

### 2.3. Statistical Analysis

Concerning data collection, the anonymisation was ensured. Patient characteristics were presented as the mean with standard deviation (±SD) regarding age and frequency (%) for gender and clinical characteristics of the cancer.

Continuous variables were assessed using *t*-tests when the normality of distribution was verified; otherwise, Mann–Whitney U tests were applied for non-normally distributed data. Categorical variables were evaluated using chi-square tests. A *p*-value ≤ 0.05 was considered the threshold for statistical significance. Statistical analyses were performed by using the Statistical Package for Social Sciences (SPSS) version 23 (IBM Corp., Armonk, NY, USA) on a Windows 11 platform.

### 2.4. Ethical Approval

This study was approved by the institutional Pulmonology and Nutritional Departments, Ethics Commission (nº 62/CECHL/2023), and the Administration Council of the Centro Hospitalar de Leiria at 9 August 2023, where this study was conducted.

This study respects the principles of the Declaration of Helsinki, ensuring maximum protection and confidentiality of the data obtained by the participants.

## 3. Results

### 3.1. Cohort Characterisation

This cohort included all patients hospitalised with LC between 2022 and 2023 and referred to nutritional evaluation. A total of 53 patients were included in our study for analysis, primarily males (38; 71.7%) and females (5; 28.3%) with a mean age of 64 ± 13 (Table 1). Most patients presented with an Eastern Cooperative Oncology Group (ECOG) Score of 1 (47.2%) (Table 2).

### 3.2. Types of Lung Cancer

Lung adenocarcinoma was the most common cancer: it was diagnosed in 32 patients (60.34%). Most patients (45; 84.9%) presented with cancer in an advanced stage (stage IV).

### 3.3. Hospitalisation

Patients were hospitalised due to multiple reasons; however, the most common reason was pneumonia (18; 34%), followed by cancer progression (6; 11.3%) and uncontrolled pain (n = 5; 9.4%). The mean hospitalisation was 16 ± 9 days, with a minimum of 2 and a maximum of 40 days.

### 3.4. Nutritional Assessment

Relevant nutritional symptoms were assessed, and the most significant symptom was anorexia, which was observed in 43 (81.1%) patients (Table 3). Other symptoms included obstipation and xerostomia. Three patients had no symptoms reported.

Forty-eight patients (90.6%) were at nutritional risk at hospitalisation, defined by an NRS score ≥3. According to the GLIM criteria, 24 (45.3%) and 19 (35.8%) patients had moderate and severe malnutrition, respectively.

At the time of discharge, after the nutritional intervention, there was a statistically significant decrease in the number of patients with nutritional risk (*p* = 0.006), with a change in the NRS score to <3.

### 3.5. Nutritional Intervention

All patients had a nutritional assessment, and the level of support and interventions was adjusted individually according to the nutritional risk/status.

Nutritional interventions included dietary counselling, changes in hospital diet (texture modification, salt addition, fibre addition), food supplements (such as yoghurts), and oral nutritional supplements.

For this analysis, patients were divided into two groups for direct comparison: patients with an NRS score improvement (group 1) and patients without improvement (group 2). The results are presented in Table 4.

Both groups had male patient prevalence, with no significant age (*p* = 0.153) or ECOG differences (*p* = 0.556). At the time of hospitalisation, there was no significant difference among initial NRS scores (*p* = 0.556).

Patients with stage IV cancer were associated with severe malnutrition (*p* = 0.019). However, they had significantly improved NRS scores (*p* = 0.044).

Patients with a higher nutritional intake also had improved NRS scores (*p* = 0.04).

A total of 26 patients (49.1%) were treated with artificial nutritional support, but no significant impact was observed in the NRS scores (*p* = 0.328).

## 4. Discussion

This study aimed to assess the risk and nutritional status of LC patients who are hospitalised, as well as to evaluate the impact of nutritional intervention on the risk of malnutrition. Most patients were conscious, collaborative, and autonomous, which is considered an added value for data collection, as it allowed for obtaining data more reliably and was also a facilitating element with regard to nutritional intervention.

From the analysis of the characteristics of the studied sample, it is worth mentioning that the majority of patients included in this sample are male, 38 (71.7%), which agrees with most studies on this type of cancer [1,2].

LC patients present a high frequency of malnutrition at the time of hospitalisation; the proportion of patients at nutritional risk at the beginning of hospitalisation found in this study aligns with what is reported in the literature for oncology patients [4,5].

All available literature emphasises that investments in nutritional support yield economic returns and improvements in the patient’s quality of life [3,4,6,10]. The NRS2002 was used as a form of screening, as it is easy to use, quick to apply, and validated for the population focused on in this study (inpatient hospital) [6,7,12].

Nutritional status was only classified at the beginning of hospitalisation due to hospital service logistics. The literature supports using the GLIM in diagnosing malnutrition and predicting survival among LC patients [22,23,24].

According to the literature, insufficient food intake increases the prevalence and severity of malnutrition, with a concomitant increase in the associated complications, which is why every hospitalised patient should have a nutritional care plan that identifies their individual dietary needs and the corresponding treatment plan. Personalised nutritional interventions are key in helping to manage potential side effects of illness and treatments, such as anorexia and dysphagia. These data show that nutritional intervention for all patients has beneficial effects associated with reducing patients’ nutritional risk [3,13,15,16].

Nutritional interventions can help patients maintain body weight by increasing food intake and by providing education on dietary strategies such as meals and snacks. Small, frequent meals and snacks can be helpful if appetite or intake is poor [9,15,16,17].

The significant positive aspect of this study is the reinforcement and validation of the multiple benefits a personalised nutritional intervention has to patients with LC, despite being hospitalised and in an advanced stage of the disease, by literature reports [13,15,20].

The length of stay did not influence the results of the NRS2002, which is why nutritional intervention must be considered regardless of the length of stay, and ideally from the beginning of hospitalisation [4,8,18,19].

To limit possible bias and inter-observer variation, all the data were collected by the same principal two investigators responsible for this study, and patients were selected from the same hospital department.

The selected cohort represents a specific group of patients hospitalised with LC. Therefore, it must fully describe the benefits of nutritional evaluation and support to patients with LC or others.

The nature of this research, which is not an experimental study, is a limitation. However, the randomisation of two groups, providing nutritional support to one group and none to the other, to better conclude the benefits of this intervention would certainly impose serious ethical issues. Other biases in this study were the variable length of hospitalisation, causes of hospitalisation, and nutritional intervention. Also, multiple types and stages of cancer were included. Some patients were referred to nutritional support in the early stages, but others were referred later in the course of the disease.

Although we have only measured the impact of nutritional intervention by changes in nutritional risk, it is known that nutritional intervention in hospitalised lung cancer patients can offer several advantages [13,14,21,26]:Improvement of nutritional status: adequate nutrition can help correct nutritional deficiencies common in cancer patients due to factors such as loss of appetite and side effects of treatments.Increase muscle strength: proper nutrition can help preserve muscle mass, which is crucial for recovery and the patient’s functional capacity.Improvement in treatment tolerance: an adequate nutritional state can enhance tolerance to treatments such as chemotherapy and radiotherapy, reducing side effects.Support for the immune system: essential nutrients can strengthen the immune system, helping the body to combat infections and other complications.Quality of life: good nutrition can improve quality of life, alleviating symptoms such as fatigue and weakness.Reduction in complications: nutritional intervention can help prevent malnutrition-related complications, such as infections and respiratory issues.Individualisation of treatment: nutritional assessment allows for the personalisation of diet, tailoring it to the patient’s specific needs while considering their clinical status and preferences.Psychological support: nutrition can positively impact mental health, helping to enhance the patient’s emotional well-being.

These advantages highlight the importance of a multidisciplinary approach in treating lung cancer patients, where nutrition plays a fundamental role in recovery and disease management [4,6,16,27,28,29].

With the findings of this study, the authors intend to promote early nutritional interventions in all hospitalised patients with lung cancer, even with advanced disease, and sensitise all healthcare professionals to the relevance of this therapeutic approach [28,29,30]. More studies with larger populations are required to validate other benefits, such as mortality and prognosis impact [4,6].

## 5. Conclusions

Most patients hospitalised with LC exhibit an altered nutritional status. Prompt nutritional screening using the NRS-2002 is strongly advised. If there is a risk of malnutrition, the patient should be swiftly referred to a clinical nutritionist for further assessment and nutritional therapy. In our study, the personalised nutritional intervention reduced the proportion of patients at nutritional risk, which is crucial for LC patients to mitigate the adverse effects of malnutrition in this population. Furthermore, based on our daily clinical experience, clinical nutritionists should be involved in tumour boards to provide nutritional support to lung cancer patients when necessary. Nutritional intervention should start with precise counselling and a personalised diet at the time of the diagnosis. However, more extensive studies are required to ascertain the prognostic impact of nutritional interventions.

## Figures and Tables

**Table 1 nutrients-17-00006-t001:** Subject characterisation.

Variables	Results *n* (%)
Total of patients	53 (100)
Male patients	38 (71.7)
Age, mean years (SD)	64 (13)
Minimal age	39
Maximal age	89

Abbreviations: SD—standard deviation.

**Table 2 nutrients-17-00006-t002:** Clinical characteristics of cancer.

Variables	Results *n* (%)
ECOG Score, *n* (%)	
0	0 (0)
1	25 (47.2)
2	2 (3.8)
3	18 (34)
4	8 (15.1)
Types of lung cancer, *n* (%)	
Adenocarcinoma	32 (60.4)
Squamous-cell carcinoma	8 (15.1)
Small-cell carcinoma	5 (9.4)
Undetermined/unknown	4 (7.5)
Metastasis	3 (5.7)
Large-cell carcinoma	1 (1.9)
Cancer stage, *n* (%)	
I	1 (1.9)
II	2 (1.9)
III	4 (7.5)
IV	45 (84.9)
Undetermined/unknown	2 (3.8)

Abbreviations: ECOG—Eastern Cooperative Oncology Group.

**Table 3 nutrients-17-00006-t003:** Nutritional symptoms and characterisation of the nutritional status.

Variables	Results (t_0_) *n* (%)	Results (t_1_) *n* (%)	*p*-Value
Major nutritional symptoms			
Anorexia	43 (81.1)		
Dysphagia	4 (7.5)		
Other symptoms	3 (5.7)		
No symptoms	3 (5.7)		
NRS score, *n* (%)			
0	2 (3.8)	1 (1.9)	
1	0 (0)	1 (1.9)	
2	3 (5.7)	12 (22.6)	
3	26 (49.1)	21 (39.6)	
4	17 (32.1)	15 (28.3)	
5	4 (7.5)	2 (3.8)	
6	1 (1.9)	1 (1.9)	
7	0 (0)	0 (0)	
Nutritional risk (NRS ≥ 3), *n* (%)	48 (90.6)	39 (73.6)	0.006
GLIM criteria, *n* (%)			
Nutritional risk	10 (18.9)		
Moderate malnutrition	24 (45.3)		
Severe malnutrition	19 (35.8)		

Abbreviations: t_0_—at hospitalisation; t_1_—at discharge; NRS—Nutrition Risk Screening; GLIM—Global Leadership Initiative on Malnutrition.

**Table 4 nutrients-17-00006-t004:** Characteristics of patients with Nutrition Risk Screening improvement.

Variables	Group 1 *n* (%)	Group 2 *n* (%)	*p*-Value
Total of patients	17 (100)	36 (100)	
Male patients	11 (64.7)	27 (75)	0.520
Female patients	6 (35.3)	9 (25)	
Age, median years (IQR)	61 (19)	66 (19)	0.153
Days of hospitalisation, median days (IQR)	12 (10)	15 (3)	0.335
ECOG, median (IQR)	2 (3)	3 (2)	0.556
Stage IV cancer, *n* (%)	17 (100)	28 (77.8)	0.044
NRS score t_0_, median (IQR)	3 (1)	3 (1)	0.438
Severe malnutrition, *n* (%)	3 (17.6)	16 (44.4)	0.058
Nutritional intake, median %, (IQR)	75 (25)	50 (25)	0.040
Artificial nutrition, *n* (%)	10 (58.8)	16 (44.4)	0.328

Abbreviations: IQR—interquartile range.

## Data Availability

Data are contained within this article.

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
