# Peer review of "The Effect of Nutritional Intervention in Nutritional Risk Screening on Hospitalised Lung Cancer Patients"

_nutrients, 2024, doi:10.3390/nu17010006_

Round 1
Reviewer 1 Report
Comments and Suggestions for Authors
Title: Nutritional intervention in hospitalized lung cancer patients – what impact on nutritional risk screening?
Lung cancer is the most common cancer worldwide and the leading cause of death in men. This review suggests that nutritional interventions should be implemented to enhance prognosis and improve patients' health outcomes.
Reviewer comments and suggestions
1. I suggest revising the title: "what impact does nutritional risk screening have?” What do the authors mean?"
2. parenteral nutrition.16 Dietary – is the number 16 a reference? – line 107
3. The study was financed in the Centro Hospitalar de Leiria between January 2022 and December 2023 – do the authors mean “conducted”? – lined 126-127
4. I suggest rewriting this statement; it is not clear – “as well as biometric parameters evaluations, to apply Nutritional Risk Screening 2002 (NRS2002) (at the beginning and the end of hospital stay) and to determine nutritional status by the Global Leadership Initiative on Malnutrition (Glim) criteria [22-24] – lines 137-139
5. Please add the number of females- “A total of 53 patients were included in our study for analysis, primarily males (38; 71.7%),” – lines 169-170
6. Please revise the statement to clarify whether the authors mean "48 patients (90.6%)" or "patient 48." I assume that the patients were de-identified using codes ranging from Patient 01 to Patient 53. Please provide clarification. – Line 189
7. Please add the values for the standard deviation (SD) for Tables 1 through 3. The reader cannot find the SD in the tables; it is listed in the legend.
8. Please add the values for female patients in Table 4
9. Days of hospitalization, median days (IQR) 12 (10) 15 (3) 0,335 – please add a period instead of a comma
10. Since the study's goal was to provide evidence for individualizing nutritional support to LC patients, the author should include more literature on this topic in the discussion.
Comments on the Quality of English LanguageIt must be improved
Author Response
- I suggest revising the title: "what impact does nutritional risk screening have?” What do the authors mean?"
Changed: The Effect of Nutritional Intervention in Nutritional Risk Screening on Hospitalized Lung Cancer Patients
- parenteral nutrition.16 Dietary – is the number 16 a reference? – line 107
Changed.
- The study was financed in the Centro Hospitalar de Leiria between January 2022 and December 2023 – do the authors mean “conducted”? – lined 126-127
Changed.
- I suggest rewriting this statement; it is not clear – “as well as biometric parameters evaluations, to apply Nutritional Risk Screening 2002 (NRS2002) (at the beginning and the end of hospital stay) and to determine nutritional status by the Global Leadership Initiative on Malnutrition (Glim) criteria [22-24] – lines 137-139
Changed.
- Please add the number of females- “A total of 53 patients were included in our study for analysis - males N=38; 71.7% and females N=15; 28.3%” – lines 169-170
Changed.
- Please revise the statement to clarify whether the authors mean "48 patients (90.6%)" or "patient 48." I assume that the patients were de-identified using codes ranging from Patient 01 to Patient 53. Please provide clarification. – Line 189
Means 48 of patients; in not related to the “number” of the patient.
- Please add the values for the standard deviation (SD) for Tables 1 through 3. The reader cannot find the SD in the tables; it is listed in the legend.
Apenas foi apresentada media e desvio padrão na tabela 1; na tabela 2 e 3 a legenda “SD” foi copy past da tabela 1 e já foi corrigido.
- Please add the values for female patients in Table 4
Included.
- Days of hospitalization, median days (IQR) 12 (10) 15 (3) 0,335 – please add a period instead of a comma
Changed.
- Since the study's goal was to provide evidence for individualizing nutritional support to LC patients, the author should include more literature on this topic in the discussion.
More references were included in discussion.
Reviewer 2 Report
Comments and Suggestions for Authors
Dear Editor and Authors,
I have evaluated this research paper titled “Nutritional intervention in hospitalized lung cancer patients what impact on nutritional risk screening?” by Dr. Oliveira and her collegues from Faro and Coimbra, Portugal.
This is a single institution, small, retrospective, observational study in which the authors have assessed the nutritional status and risk for malnutrition in 53 patients with lung cancer and have implemented nutritional interventions including counseling and caloric increase via diet modification and/or nutritional supplementation. They showed a significant improvement in nutritional risk from 90.6% to 73.6% at the time of discharge.
Overall this in not really neither a novel concept nor a pioneering approach to cancer patient management! In this reviewer’s for example department, all lung cancer patients (who are also surgical patients) undergo a nutritional risk assessment and are placed on hi-caloric/hi-protein diet as needed! Similarly, there has been extensive work done and literature published on the matter.
I have the following comments:
1. The introduction is quite comprehensive and gives a good overview of the problem and the aims of the study.
2. How could this be a retrospective study? Where all lung cancer patients routinely given a nutritional assessment on admission (similar to what is mentioned above)? Why wasn’t a power analysis performed to evaluate if the sample is powerful enough to produce statistically meaningful results?
3. If the weight was “guestimated” for some patients as the authors have reported on line 144 how was bias been prevented? i.e. underscoring the patient’s weight either consciously or unconsciously.
4. How was it possible to assess patient food intake from the patient’s chart!! There only prescribed diet is recorded with no measure how much of the allocated food the patient actually consumed!! A plate can be put in front of a patient but you can’t force him to eat it!!
5. The statistical analysis section needs to be re-written in more accurate and scientific wording! Why wasn’t a multivariate analysis performed to assess risk factors and variables?
6. In line 168 the authors “let it slip” that these 53 patients where the ones which were referred for nutritional assessment (presumably by the nursing/medical staff) because they were clearly at nutritional risk. As such it is not unlikely that they represent the more affected group where small interventions could produce significant improvement! This is certainly another bias!!
7. How many patients died during the study? Where there any patients that where assessed and subsequently died so they were removed from the final number?
8. The language of the manuscript is adequate and needs only minor editing!
In conclusion, this is not a novel and exciting study! It is based on either already known and accepted concepts which are widely assessed and modified as part of routine clinical practice. Nevertheless, despite its relatively reduced clinical significance this study does reinforce the already established concepts and could potentially be of some limited benefit, provided my comments are addressed in the revision. Thank you and I am awaiting your revised work.
Author Response
- How could this be a retrospective study? Where all lung cancer patients routinely given a nutritional assessment on admission (similar to what is mentioned above)? Why wasn’t a power analysis performed to evaluate if the sample is powerful enough to produce statistically meaningful results?
Thank you for your comment. We completely agree and in fact it was a mistake. It’s a prospective study. We corrected in the manuscript.
An analysis of the statistical power of the sample was not performed because all patients with lung cancer hospitalized during the period covered by the analysis were included in the present study.
- If the weight was “guestimated” for some patients as the authors have reported on line 144 how was bias been prevented? i.e. underscoring the patient’s weight either consciously or unconsciously.
Thank you for your comment. The weight estimate was made using the formula by Chumlea et al. (1988), which is why no potential observer bias associated with the estimate is identified. The methodology was updated to include a reference to the weight estimate, as described above.
- How was it possible to assess patient food intake from the patient’s chart!! There only prescribed diet is recorded with no measure how much of the allocated food the patient actually consumed!! A plate can be put in front of a patient but you can’t force him to eat it!!
We thank for the reviewer comment. In the hospital (and in all hospitals in Portugal), nursing team after training for dietitians, collected in each meal the percentage of intake for patient, individually. This information is added in the patients’ medical file and could be consulted by dietitians, physicians and other healthcare professionals.
- The statistical analysis section needs to be re-written in more accurate and scientific wording! Why wasn’t a multivariate analysis performed to assess risk factors and variables?
We thank for the reviewer comment. We re-write the statistical analysis section. We hope it is now more scientific. In this paper the main objective was to evaluate the impact of nutritional intervention in nutritional risk, reason why the authors didn’t perform the multivariate analysis.
- In line 168 the authors “let it slip” that these 53 patients where the ones which were referred for nutritional assessment (presumably by the nursing/medical staff) because they were clearly at nutritional risk. As such it is not unlikely that they represent the more affected group where small interventions could produce significant improvement! This is certainly another bias!!
We thank for the reviewer comment. All the patients with lung cancer in this hospital were included in the study, reason why we have 5 patients (about 10% of the sample) without nutritional risk (NRS 2002 < 3).
- How many patients died during the study? Where there any patients that where assessed and subsequently died so they were removed from the final number?
We thank for the reviewer comment. During the study, 15 patients died. However any patient was removed from the final number, because it was considered the last Nutritional Risk Score evaluated before dead (the evaluation were performed weekly).
- The language of the manuscript is adequate and needs only minor editing!
In conclusion, this is not a novel and exciting study! It is based on either already known and accepted concepts which are widely assessed and modified as part of routine clinical practice. Nevertheless, despite its relatively reduced clinical significance this study does reinforce the already established concepts and could potentially be of some limited benefit, provided my comments are addressed in the revision. Thank you and I am awaiting your revised work.